


# Assessment of Pollution-Health-Economics Nexus in China

Yang Xia[1], Dabo Guan[1*], Jing Meng[2], Yuan Li[1], Yuli Shan[1]

[1] Water Security Research Centre, School of International Development, University of East Anglia, Norwich NR4 7TJ, UK

[2] Department of Politics and International Studies, University of Cambridge, Cambridge CB3 9DT, UK

* Correspondence email: Dabo.Guan@uea.ac.uk

## Abstract

Serious haze can cause contaminant diseases that trigger productive labour time by raising

mortality and morbidity rates in cardiovascular and respiratory diseases. Health studies rarely consider macroeconomic impacts of industrial interlinkages while disaster studies seldom involve air pollution and its health consequences. This study adopts a supply-driven input-output model to estimate the economic loss resulting from disease-induced working time reduction across 30 Chinese provinces in 2012 using the most updated Chinese Multiregional Input-Output Table.

Results show total economic loss of 398.23 billion Yuan (~1% of China's GDP in 2012) with the majority comes from Eastern China and Mid-South. Total number of affected labourers amounts at 82.19 million. Cross-regional economic impact analysis indicates that Mid-South, North China and Eastern China entail the majority of regional indirect loss. Indeed, most indirect loss in North China, Northwest and Southwest can be attributed to Manufacturing and Energy in other regions

while loss in Eastern China, Mid-South and Northeast largely originate from Coal and Mining in other regions. At the sub-industrial level, most inner-regional loss in North China and Northwest originate from Coal and Mining, in Eastern China and Southwest from Equipment and Energy, and in Mid-South from Metal and Non-metal. These findings highlight the potential role of geographical distance in regional interlinkages and regional heterogeneity in inner- and

outer-regional loss due to distinctive regional economic structures and dependences between the North and South.

Keywords: China, 2012, PM$_{2.5}$ air pollution, Indirect economic loss, Health, Input-Output analysis






# 1. Introduction

Millions of people in China are currently breathing a toxic cocktail of chemicals, which has become one of the most serious topics in environmental issues in China by resulting in
widespread environmental and health problems, including increasing risks for heart and respiratory diseases, stroke and lung cancer (LC) (Greenpeace, 2017). As air pollution has long-term health impacts that evolves gradually over time, understanding the health and socioeconomic impacts of China's air pollution requires continuous efforts.

Serious air pollution in China has largely inspired epidemic studies that examine specific health outcomes from air pollution, as well as health costs assessments that translate health outcomes into monetary loss. Existing epidemic studies simulate a exposure-response relationships between Particulate Matter (PM) concentration levels and relative risks (RRs) for a particular disease while health costs assessments frequently stem from patients' perspective at
microeconomic level, by evaluating either their willingness-to-pay (WTP) for avoiding disease risk or the potentially productive years of life loss (PPYLL). However, when perceiving unhealthy laborers as degradation in labor input, macroeconomic implications for production supply chains lack investigation. Meanwhile, as the health effects of air pollution are built up slowly over time which implies the lasting nature of air pollution, it has been rarely studied in current disaster risk
literature. Differing from rapid-onset disaster analysis (flood, hurricane, earthquake, etc) that normally reply on quantifying damages to physical capital, air pollution affect more human capital than physical capital and the resulting health impacts are relatively invisible and unmeasurable. As a result, linking PM concentrations with health endpoints and further with macroeconomic impacts require an interdisciplinary approach that integrates all the three elements into one.
Inspired by our previous work on the socioeconomic impacts of China's air pollution in 2007, this paper applies the similar approach to China's air pollution in 2012 and also examines the cross-regional economic impacts in order to underline the important role of indirect economic loss. Given that the majority of economic loss originate from secondary industry, this paper also specifically analyze the key sectors in secondary industry that account for the greatest
proportions in both direct and indirect economic loss in each great region in China. By doing so, future policymakers and researchers could obtain an alternative macroeconomic tool to better conduct cost-benefit analysis in any environmental or climate change related policy design, and to comprehend health costs studies in its macroeconomic side.

# 2. Results

## 2.1 Total Number of Affected Labor and Total Economic Loss

Firstly, regarding the total number of affected labour and total economic loss, the total economic loss resulting from $PM_{2.5}$-induced health outcomes in China 2012 is 398.23 billion Yuan, which corresponds to almost 1% of national GDP in 2012. The total number of affected labour in China is 0.80 million for $PM_{2.5}$-induced mortality, 2.22 million for $PM_{2.5}$-induced hospital admissions and
79.17 million for $PM_{2.5}$-induced outpatient visits (*Fig.1*). *Figure 1* presents the provincial counts of $PM_{2.5}$-induced mortality, hospital admissions, outpatient visits and economic loss with least severe and most severe situation shown from green to red. For total populations of




PM$_{2.5}$-induced mortality and morbidity, among 30 provinces, Henan and Shangdong province have the largest total counts of PM$_{2.5}$-induced mortality and morbidity, which is consistent with

the findings in 2007 study (Xia et al, 2016). Guangdong province has the greatest counts of PM$_{2.5}$-induced hospital admissions at 291 thousands, where a substantial increase can be observed at 175 thousands compared with results in 2007. It almost doubles its provincial count of outpatient visits and triples its mortality counts. Meanwhile, increase can be observed in both counts for Northwest region, including Shanxi, Gansu, Qinghai, Ningxia and Xinjiang provinces.

Specifically, the count of hospital admissions in Shanxi province in 2012, 100 thousands, also doubles that of 2007, which was at 50 thousands. Even sharper increase of admission counts can be seen in Xinjiang province, where the number is almost 7 times of that from 2007.

*fig01. Provincial counts of PM$_{2.5}$-induced mortality, hospital Admissions, outpatient visits and*

*economic loss in China, 2012*

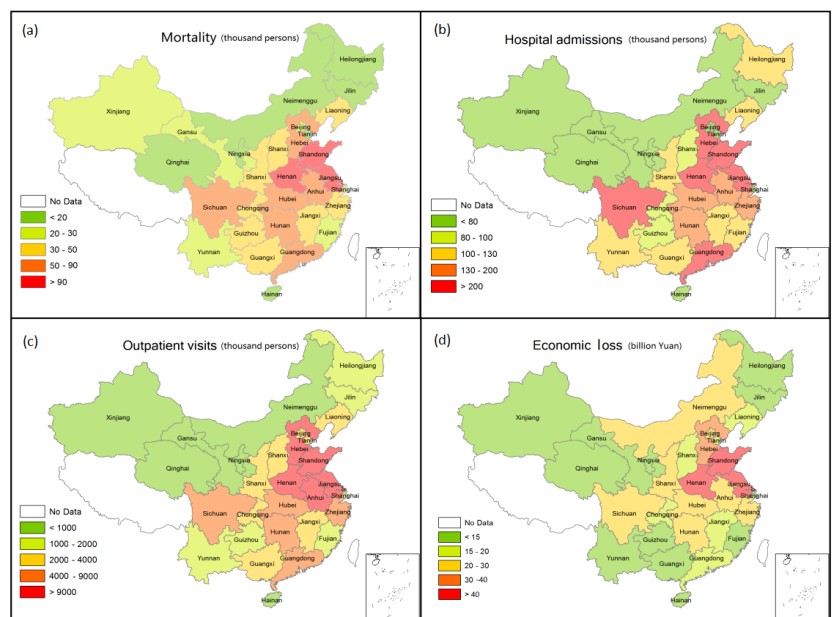

*Provincial counts of PM$_{2.5}$-induced mortality (a), hospital admissions (b), outpatient visits (c) and economic loss (d) are displayed in four panels above, with least severe and most severe situation shown from green to red. We did not consider Tibet due to the lack of data.*

## 2.2 Economic Loss by Provinces, Regions and Industries


Secondly, concerning economic loss by provinces, regions and industries, at the provincial level (*Fig.1*), economic loss of Henan province exceeds that of Jiangsu province in 2007 (55.90 billion Yuan) and becomes the province suffering the greatest economic loss at 56.37 billion, accounting for 14% of the total economic loss in China. This is followed by Jiangsu province at 45.32 billion

Yuan and Shangdong province at 43.23 billion Yuan. This is because all the three provinces have the largest counts of PM$_{2.5}$-induced mortality and morbidity, which result in substantial provincial labour time loss. I also calculated the economic loss in six China's great regions. Eastern China



and Mid-South appear to be the two regions suffering the greatest economic loss that amount at 153.39 and 119.21 billion Yuan and account for 39% and 30% of total economic loss in China,
2012. It is in line with the findings from 2007 study (Xia et al, 2016), where the economic loss of these two regions are 115.33 and 80.88 billion Yuan respectively. Therefore, there has been a remarkable rise in economic loss for Mid-South region. Primary industry includes agriculture and fishing suffered the economic loss at 19.12 billion Yuan. Secondary industry includes all manufacturing sectors, energy and construction and it entails the greatest proportion of
economic loss at 320.06 billion Yuan (80% of total economic loss). Tertiary industry accounts for the remaining 15% of total economic loss at 59.05 billion Yuan.

## 2.3 Cross-regional Economic Loss

Additionally, this case study also examined the cross-regional economic losses between six Great Regions in China. As one significant advantage for input-output model is to capture the industrial
and regional interdependencies, it is effective to measure the propagating disaster-induced indirect economic loss along production supply chain. I traced the cross-regional economic loss due to their interlinkages, such as interregional trade, as shown in *Fig.2*. The diagram demonstrates the interregional economic impacts due to their interdependencies. The left-hand side shows the regional indirect economic loss while the right-hand side denotes the sources for
these indirect economic loss. The proportion of regional indirect loss among regional total economic loss is displayed next to each region's name on the left-hand side. Although the majority of regional economic loss come from the direct economic loss occurred within the region across almost all the six regions, Northeast, Eastern China and Northwest still entail great indirect economic loss from other regions, which occupies 31%, 21% and 30% of the total
regional economic loss, respectively. In Northeast, a totality of 18% of its total regional economic loss is originated from North China and Mid-South, including 1.84 billion Yuan from North China and 1.85 billion Yuan from Mid-South. Similarly, Mid-South is responsible for 9% of the economic loss in Eastern China at 13.36 billion Yuan. It accounts for even larger proportion of regional economic loss in Northwest at 13%. Meanwhile, Eastern China also accounts for another 8% of
the total regional economic loss in Northeast, which amounts at 1.66 billion Yuan. Overall, Mid-South accounts for the largest amount of indirect economic loss in other Chinese regions at 24.65 billion Yuan, which is followed by North China and Eastern China at 16.99 and 12.17 billion Yuan, respectively. This finding highlights the increasing significance in capturing the industrial and regional interdependencies and indirect economic loss in disaster risk analysis because such
interdependencies can largely raise the overall economic loss far beyond the direct economic loss and constitute a noticeable component of total economic loss.





*fig02. Cross-regional economic loss analysis*

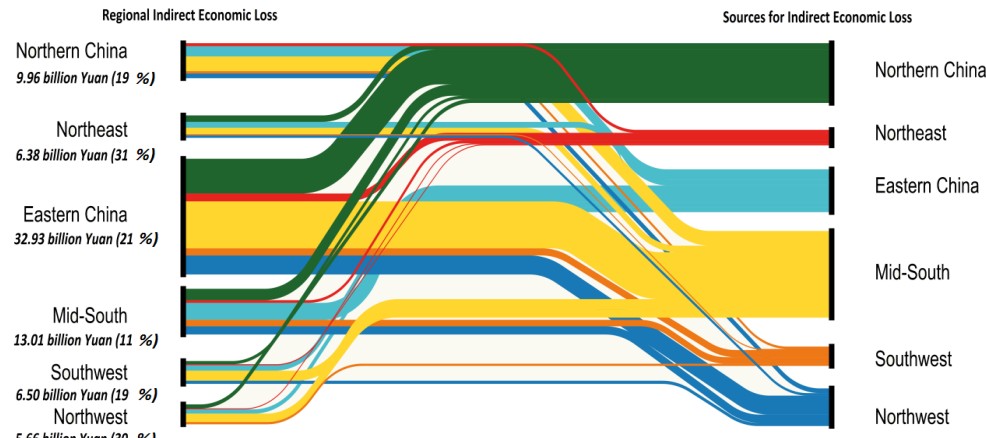

*The diagram demonstrates the interregional economic impacts due to their interdependencies.*
*The left-hand side shows the regional indirect economic loss while the right-hand side denotes the*
*sources for these indirect economic losses. The proportion of regional indirect loss among*
*regional total economic loss is displayed next to each region's name on the left-hand side.*

## 2.4 Regional Direct and Indirect Loss from Secondary Sector

As secondary sector plays a vital role in Chinese economy and entails greatest economic loss among the three industries, I specifically analyzed the regional economic loss that are directly and indirectly resulting from secondary sectors both inside and outside a region. Focusing on secondary sector, *Figure 3* illustrates both direct and indirect economic loss originating from each region and outside the region. As can be seen from the diagram, despite that the majority of economic loss resulting from secondary sector are originated from inside the region for all the six great regions in China, in Northwest and Northeast, economic loss attributed to secondary sectors outside the region still constitute a considerable share due to industrial and regional interdependencies. Secondary sectors in the Mid-South, Eastern China and North China become three major sources for indirect economic loss across all the six regions. For instances, in Northwest, economic loss from secondary sectors in Mid-South, Eastern China and North China account for 14%, 6% and 6% of total regional indirect loss from secondary sectors outside the region, at 2.20, 0.99 and 0.90 billion Yuan, respectively. Similarly, in Northeast, economic loss from secondary sectors in these three regions occupy 10%, 8% and 9% of total regional indirect loss from secondary sectors outside the region, at 1.66, 1.33 and 1.46 billion Yuan, respectively. This is resulting from their geographical distance to Mid-South, Eastern China and North China, as well as close trade relationships with these three regions. The significant roles of Mid-South and Eastern China in interregional trade have been early confirmed by Sun and Peng (2011), where they pointed out the export-oriented nature for trades in Eastern China and Mid-South, and their close trade relations with Northwest regions with respects to imports of raw materials. Likewise, it is noticeable that indirect economic loss is more likely to come from neighbour-regions, which highlights the possibility that short geographical distance might accelerate interregional trade and strengthen regional interlinkages.



*fig03. Regional direct and indirect economic loss from secondary sectors*

[figure]

*The inner ring denotes the direct economic loss originating from secondary sector inside the region while the outer ring stands for the indirect economic loss from secondary sectors in other regions. Percentage shown on the inner ring shows the proportion of direct economic loss regarding total regional economic loss and percentages shown on the outer ring are the proportions of indirect loss from other regions relative to total regional indirect economic loss.*

## 2.5 Direct, Indirect Loss from Subindustries in Secondary Sector

Secondary sector was further broken down into seven industries in order to examine the major economic loss sources among subindustries of secondary sectors inside and outside the region. They include Coal and Mining, Manufacturing, Fuel processing and Chemicals, Metal and Non-metal, Equipments, Energy and Constructions as displayed in *Fig.4*. In North China, Northwest and Southwest, most of their indirect economic loss from secondary sectors outside the region comes from Manufacturing with 27.0%, 26.7% and 22.2%, respectively. The second largest source in these three regions that accounts for economic loss from secondary sectors in other regions is Energy, with the greatest amount occurs in North China at 2.32 billion Yuan, followed by Northwest at 1.29 billion Yuan and Southwest at 1.26 billion Yuan. In contrast, Coal and Mining accounts for the majority of indirect loss from secondary sectors outside the region for Eastern China, Mid-South and Northeast at 37.4% (10.83 billion Yuan), 33.4% (3.65 billion Yuan) and 24.4% (1.30 billion Yuan), respectively. One possible underlying reason is that economies in Northwest, North China and Southwest are mainly dominated by Coal and Mining but relying on imports of Manufacturing products from other regions, whereas Eastern China, Mid-South and Northeast have more prosperous Manufacturing industries but tend to heavily depend on imports of raw materials from Coal and Mining industries in Northwest, North China or Southwest. With regards to the economic loss from secondary sector inside each region, it



shows diversified patterns across six great regions. Coal and Mining accounts for the largest part of inner-regional economic loss in North China and Northwest at 42.4% and 43.8%, respectively, Equipments and Energy appear to be two major sources for inner-regional economic loss Eastern
China and Southwest, while Metal and Non-metal and Manufacturing constitute considerable proportions in inner-regional economic loss from secondary sectors in Mid-South, which reach 21.86 billion Yuan and 21.61 billion Yuan, occupying 27.4% and 27.1%, respectively.

*fig04. Economic loss from seven industries in secondary sector inside and outside the region*




*The inner circle shows the economic loss from secondary sector inside the region. The size of circle stands for the different proportions of inner-regional economic loss relative to total regional economic loss. Colors demonstrate economic loss from seven sectors in secondary sector inside*
*the region. Meanwhile, the outer circle indicates the economic loss from secondary sectors outside the region. Economic loss resulting from seven sectors are shown in black and white. Percentages shown on the outer circle are the proportions of indirect loss from other regions relative to total regional indirect economic loss.*

# 3. Discussions

PM$_{2.5}$ has seriously undermined human health by inducing contaminant diseases, including IHD, Stroke, COPD and LC. These diseases have resulted in substantial numbers of mortality and morbidity that further cause labor degradation in terms of productive working time loss along production supply chain. Therefore, there is a growing need to explore the macroeconomic implications of PM$_{2.5}$-induced health effects that can also capture industrial and regional
interdependencies. However, existing health costs studies assess the health costs at microeconomic level without an investigation over these linkages on the production supply-side. Meanwhile, disaster risk studies rarely involve PM$_{2.5}$ pollution as a disaster that harm human capital more than physical capital. Thus, methods to quantify the direct damages to infrastructure seem to be inefficacious when measuring the 'damages' to human health. Inspired by the



previous study on China's air pollution in 2007 (Xia et al, 2016), the current study applies the interdisciplinary approach to assess the macroeconomic impacts of $PM_{2.5}$-induced health effects in China 2012 by perceiving reducing labor time as an indicator for reducing value added so that it can be fed back into a supply-driven IO model and health studies can be integrated into impact evaluation and interdependency analysis. The current case study applies the interdisciplinary

approach by combining environmental, epidemiological and macroeconomic studies to assess the macroeconomic impacts of $PM_{2.5}$-induced health effects in China during 2012. In the model, environmental phenomenon was related with health endpoints using an integrated exposure-response model, reduction in labour time were estimated based on the pollution-induced mortality and morbidity counts, and industrial reducing labour time was

perceived as an indicator for industrial reducing value added, which was further fed back into a supply-driven input-output model. By doing so, health studies can be integrated into impact evaluation and interdependency analysis.

The results are threefold. Firstly, the total economic loss from China's air pollution during 2012

amount at 398.23 billion Yuan with the majority comes from Eastern China (39%) and Mid-South (30%). The total economic loss is equivalent with 1.0% of China's GDP in 2012 and the total number of affected labourers rises to 82.19 million. Compared with study in 2007 (Xia et al, 2016), although secondary industry remains the industry encountering the most economic loss (80%), changes can be noticed for economic loss at provincial level. Henan and Jiangsu become

two provinces that suffering the greatest economic loss at 56.37 and 45.32 billion Yuan respectively, followed by Shangdong province with total economic loss at 43.23 billion Yuan. Henan and Shangdong provinces also have the largest numbers of $PM_{2.5}$-induced mortality, hospital admissions and outpatient visits. Secondly, the study highlights the cascading indirect economic loss triggered by industrial and regional interdependencies in health costs assessment.

In 2012, indirect economic loss constitutes a significant part of total regional economic loss in Northeast, Eastern China and Northwest, which occupies 31%, 21% and 30% of the total regional economic loss, respectively. Overall, Mid-South accounts for the largest amount of indirect economic loss in other Chinese regions at 24.65 billion Yuan, which is followed by North China and Eastern China at 16.99 and 12.17 billion Yuan, respectively. Additionally, the study

specifically focuses on 7 sectors in secondary industry and differentiates economic loss from these sectors inside the region from those outside the region. In Northwest and Northeast, economic loss attributed to secondary industries outside the region still constitute a considerable share due to industrial and regional interdependencies at 31% and 34% of total regional economic loss, respectively. Secondary industries in the Mid-South, Eastern China and North

China become three major sources for indirect economic loss across all the six regions. Indeed, I also suggest that indirect economic loss is more likely to come from neighbour-regions, which highlights the possibility that short geographical distance might accelerate interregional trade and strengthen regional interlinkages. In North China, Northwest and Southwest, most of their indirect economic loss are originated from Manufacturing industries outside the region with

27.0%, 26.7% and 22.2%, respectively. The second largest source in these three regions that accounts for economic loss from secondary industries in other regions is Energy, with the



greatest amount occurs in North China at 2.32 billion Yuan. In contrast, Coal and Mining accounts for the majority of indirect loss from secondary industries outside the region for Eastern China, Mid-South and Northeast at 37.4% (10.83 billion Yuan), 33.4% (3.65 billion Yuan) and 24.4% (1.30

billion Yuan), respectively. Such distinctive compositions of outer-regional economic loss might be due to the different economic structures and dependences between North China, Northwest, Southwest and Eastern China, Mid-South, Northeast. Turning to the economic loss from secondary industry inside the region, Regions show heterogeneity. Coal and Mining accounts for the largest part of inner-regional economic loss in North China and Northwest at 42.4% and

43.8%, respectively, Equipments and Energy are two major sources for inner-regional economic loss Eastern China and Southwest, while Metal and Non-metal and Manufacturing constitute considerable proportions in inner-regional economic loss from secondary industries in Mid-South.

There are some final remarks for policymakers and researchers here from this typical air

pollution issue. On the one hand, given that the prosperous interregional trade, policymakers are required to conscientiously consider these increasingly strengthened industrial and regional linkages in climate change mitigation and adaptation policy design based on a better understanding of implications resulting from any climate change-induced health issues at both micro and macroeconomic levels. Meanwhile, sufficient adaptation measures are required to be

implemented along with the climate change mitigation strategies in operation. The purpose of this is to expand the economy beyond the regional geography and natural endowment, and to release the current reliance of economy on labour-intensive sectors (Mauricio Mesquita, 2007). On the other hand, researchers on epidemic studies should actively integrate these interdependencies into future health costs evaluation while researchers on disaster risk analysis

should not lose sights on 'persistent' disasters as described in this study, which affect more human capital and may imply degradation in production factor inputs.

# 4. Methods

## 4.1 Methodological Framework






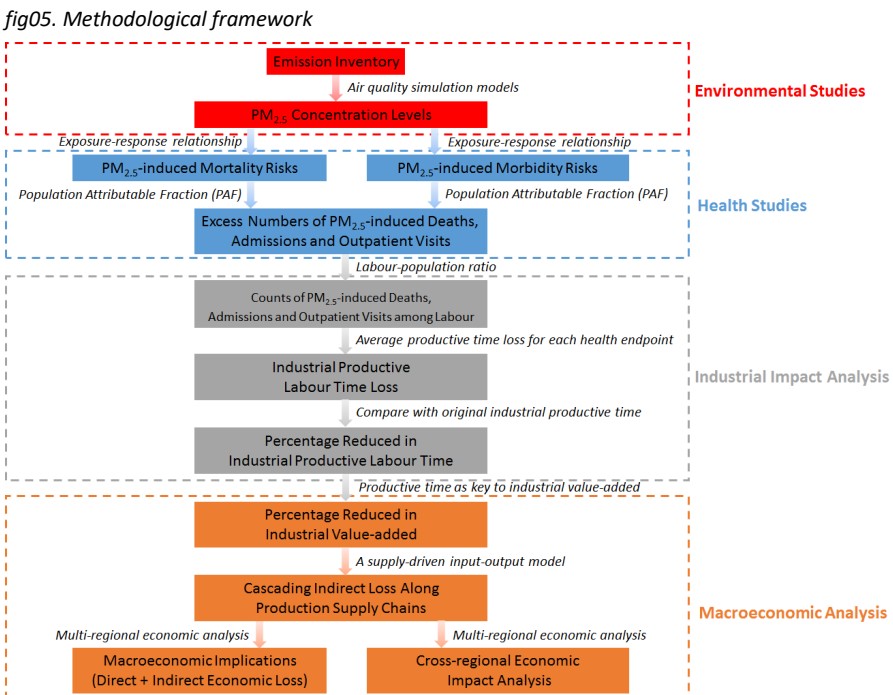

*fig05. Methodological framework*

*Figure 5 illustrates the overall methodological framework developed by this study. It involves four main parts that are distinguished with four colours. Detailed methods that connect each part in the flow chart were shown near the arrows.*


PM$_{2.5}$ concentration levels for 30 provinces of China were first identified from emission inventory using air quality simulation model. The relative risks (RRs) for PM$_{2.5}$-induced mortality (IHD, Stroke, COPD and LC), hospital admissions (cardiovascular and respiratory diseases) and

outpatient visits (all causes) were estimated using an Integrated Exposure-Response (IER) model, based on which population attributable fraction (PAF) can be calculated to estimate counts of PM$_{2.5}$-induced deaths, admissions and outpatient visits. Additionally, counts of mortality, hospital admissions and outpatient visits were further translated into productive working time loss that was compared with the industrial original working time without any PM$_{2.5}$-induced health effects

(full employment and full productivity) to derive the percentage reduction in industrial value added. Moreover, reductions in industrial value added served as an input in the supply-driven IO model to measure the total indirect economic loss incurred along the production supply chain, which is measured as the total loss in output level. Finally, macroeconomic implications regarding industrial and provincial economic loss can be obtained from our model results while

cross-regional economic impacts can be investigated through multi-regional economic analysis.

The following sections present many mathematical symbols, formulas and equations. For clarity, matrices are indicated by bold, upright capital letters (e.g., X); vectors by bold, upright lower case letters (e.g., x); and scalars by italicised lower case letters (e.g., x). Vectors are columns by

definition, so that row vectors are obtained by transposition and are indicated by a prime (e.g. **x**′ ).



A diagonal matrix with the elements of vector x on its main diagonal and all other entries equal to zero are indicated by a circumflex (e.g. $\hat{\mathbf{x}}$).

## 4.2 Provincial PM$_{2.5}$ Concentration Levels

We referred to Chinese provincial PM$_{2.5}$ concentration levels estimated by Geng et al (2015), where the authors improved the method for estimating long-term surface PM$_{2.5}$ concentrations using satellite remote sensing and a chemical transport model to assess the provincial PM$_{2.5}$ concentration levels in China during 2006-2012. The model domain includes a map of surface PM$_{2.5}$ concentrations at 0.1° × 0.1° over China using the nested-grid GEOS-Chem model with the most updated bottom-up emission inventory and satellite observations from MODIS and MISR instruments (Geng et al, 2015).

## 4.3 Health Impacts from PM$_{2.5}$ Concentration Levels

Epidemic studies on PM$_{2.5}$-induced health outcomes have linked PM$_{2.5}$ air pollution with various health endpoints by using exposure-response coefficients. This paper focuses on the impacts of PM$_{2.5}$ pollution on mortality, hospital admissions and outpatient visits. We referred to an integrated exposure-response (IER) model developed by Burnett et al (2014) to estimate the relative risks (RRs) for PM$_{2.5}$-induced mortality (IHD, Stroke, COPD, LC), hospital admissions (cardiovascular and respiratory diseases) and outpatient visits (all causes).

An IER model captures concentration-response relationships with a specific focus on ischemic heart disease (IHD), stroke, chronic obstructive pulmonary disease (COPD) and lung cancer (LC). The relative risk (RRs) for the mortality estimation function for the four diseases were shown in Eq.(1).

For $z < z_{cf}$     $RR_{IER}(z) = 1$     (Eq.1)
For $z \geq z_{cf}$     $RR_{IER}(z) = 1 + \alpha \; \{1 - \exp[-\gamma(z - z_{cf})^{\delta}] \}$

$z$: PM$_{2.5}$ exposure in micrograms per meter cubed
$z_{cf}$: counter-factual concentration level below which no additional health risk is assumed
$\delta$: the strength of PM$_{2.5}$ and $\gamma$ is the ratio of RR at low-to-high exposures

Then, the calculated RR was then converted into an attributable fraction (AF) in Eq.(2).

$$AF = \frac{RR - 1}{RR}$$     (Eq.2)

Additionally, excess counts of PM$_{2.5}$ disease-induced mortality were estimated in Eq.(3).

$$E = AF \times B \times P$$     (Eq.3)

$E$: PM$_{2.5}$-induced mortality counts, $B$ is the national level incidence of a given health effect, which was applied for all provinces because of limited data



*P*: the size of the exposed populations

380 For morbidity, we calculated cardiovascular and respiratory hospital admissions and outpatient visits for all causes using a log-linear response function and the RRs for each category of morbidity was calculated using Eq.(4).

$$RR = e^{\beta x} \tag{Eq.4}$$

385 *B*: the parameter that describes the depth of the curve (*Table SI-1* in Supplementary Information). They are the exposure-response coefficients to quantify the relationship between different levels of PM$_{2.5}$ exposures and the resulting health outcomes.

Counts of PM$_{2.5}$-induced hospital admissions and outpatient visits were analogously 390 estimated using Eq.(2) and Eq.(3).

## 4.4 Industrial Labor Time Loss

Each laborer is assumed to work 8 hours every day and 250 days during 2012. For PM$_{2.5}$-induced mortality, each death will result in a total 250 working days lost regardless different disease types. For PM$_{2.5}$-induced morbidity, each cardiovascular admission will result in 11.9 working days lost 395 while each respiratory admission causes 8.4 working days lost (National Bureau of Statistics of China, 2016). Meanwhile, we provided a range for labor time loss estimation of outpatient visits due to data unavailability, which ranges from 2 to 4 hours per outpatient visit. We assumed each outpatient visits clinic once during the year. Then, provincial mortality, hospital admissions and outpatient visits counts were scaled down to counts among labor according to labor-population 400 ratios across all the 30 provinces (Provincial Statistical Yearbook, 2013). We further distributed provincial mortality, admissions and outpatient counts into 30 industries according to industrial-total provincial labor ratio. We used industrial-total provincial output ratio instead where certain industries' labor data is missing. Additionally, labor time loss for each case of mortality, admission and outpatient visit were multiplied by industrial counts of mortality, 405 admission and outpatient in each province respectively, where the results were summed up to derive the industrial total labor time loss due to PM$_{2.5}$-induced mortality and morbidity. Moreover, we compared the industrial total labor time loss with the original labor time with full employment and labor productivity under no PM$_{2.5}$-induced health impacts. The results show the percentage reductions in industrial working time, which were used as an indicator for percentage 410 reductions in industrial value added in a supply-driven IO model as we considered labor as the major component for industrial value added.

## 4.5 Indirect Economic Loss on Production Supply Chain

We employed a supply-driven IO model to evaluate the indirect economic loss due to PM$_{2.5}$-induced mortality and morbidity along production supply chain. A supply-driven IO model 415 was developed based on a traditional Leontief IO model with the spirit of a 'circular economy'. A supply-driven IO model was derived from a traditional Leontief IO model. For a basic Leontief IO model, the total output of sector i in an *n*-sector economy can be illustrated in Eq.(5) or Eq.(6).



$$x_i = z_{i1}+....+z_{ij}+....+z_{in}+f_i = \sum_{j=1}^{n} Z_{ij} + f_i \qquad \text{(Eq.5)}$$

$$\mathbf{x = Zi + f} \qquad \text{(Eq.6)}$$

$x_i$ : the total output of sector i

$\sum_{j=1}^{n} Z_{ij}$ : the monetary value of sector i's output in all other sectors

$f_i$ : sector i's final demand that includes household final consumption, government consumption, capital formation and exports.

The basic Leontief IO model can be therefore derived in matrix notation (Eq.(7a) and Eq.(7b)).

$$\mathbf{x = Ax + f} \qquad \text{(Eq.7a)}$$
$$\mathbf{x = (I-A)^{-1}f, \ L=(I-A)^{-1}} \qquad \text{(Eq.7b)}$$

**A**: matrix of technical coefficients, $a_{ij}$, where $a_{ij} = z_{ij} / x_j$
**L**: the Leontief inverse matrix that measures the impact of value change in the final demand of a sector on the total output value on the economy (Miller and Blair, 2009).

At the same time, a supply-driven IO model takes a rotated view of Leontief IO model that shows an opposite influencing direction between sectors. It suggests that production in a sector can affect sectors purchasing its outputs as inputs during their production processes and it has a supply-side focus. A supply-driven IO model is used to calculate the impact of changes in primary
inputs on sectoral gross production. For a supply-driven IO model, the basic structure is shown in Eq.(8a) and Eq.(8b).

$$\mathbf{x' = v' \ (I-B)^{-1}} \qquad \text{(Eq.8a)}$$
$$\mathbf{x' = v' \ G, \ G = (I-B)^{-1}} \qquad \text{(Eq.8b)}$$

**B**: the allocation coefficient (direct-output coefficient), where $b_{ij} = z_{ij} / x_i$. It refers to the
440 distribution of sector i's outputs in sector j
**v**: matrix of industrial value added, including capital and labour input
**G**: the Ghosh inverse matrix, which measures the economic impacts of changes in a sector's value added on other sectors' output level

## 445 Data Availability

The data that support the findings of this study are available from the corresponding author on request.



## Author Contribution

DG and YX designed the study and YX carried them out. JM constructed the MRIO table for China 2012. YL, YS and QZ provided requested dataset. YX prepared the manuscript with contributions from all co-authors.

## Competing Interests

The authors declare that they have no conflict of interest.

## Acknowledgments

This work was supported by the National Natural Science Foundation of China (41629501), National Key R&D Program of China (2016YFA0602604 and 2016YFC0206202), National Natural Science Foundation of China (71373153, 91746112, 71773075, 71761137001 and 71503168), National Social Science Foundation of China (15ZDA054), the UK Natural Environment Research Council 460 (NE/N00714X/1 and NE/P019900/1) and Economic and Social Research Council (ES/L016028/1), British Academy Grant (AF150310) and the Philip Leverhulme Prize.



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
