# Peer review of "Assessment of Pollution-Health-Economics Nexus in China"

_Atmospheric Chemistry and Physics, 2018_

## Referee Comment (RC1) · Anonymous Referee #1 · 3 Jul 2018

This study examined the economic loss resulting from disease-induced working time reduction from PM2.5 exposure in China. The authors estimated both direct and indirect economic loss in industrial sector based on a supply-driven input-output model. Compared to their previous 2007 study, this study is superior in several aspects, including use of the most up-to-date input-output dataset of the year 2012, analysis of cross-regional impacts due to interdependence, and illustration of detailed sectoral information on economic loss in secondary industry. The manuscript is clearly written, although some sentences need to be polished (as detailed below). I would recommend publication in ACP if the authors could address the concerns I raise.

Major concerns:

1. One methodological issue is the translation of mortality into working days lost. In

section 4.4, this is described as "each death will result in a total 250 working days lost
...". It confuses me why one death only results in one-year working days lost instead of
years he supposes to work till retirement. We often consider the PM2.5-induced health
impacts in a cumulative way, and that's what the disability-adjusted life year (DALY)
stand for. DALYs represent the lost years of "healthy" life. DALYs for one premature
death is determined as the standard life expectancy at age of death in years. If one
death only accounted for one lost year of healthy life, then DALY would be significantly
underestimated. It's rather the same to calculate lost years for working. One may argue
that working days lost in years after 2012 don't account for the economic loss in 2012.
Then mortality occurring before 2012 should be taken into account to comprehend the
2012 economic loss. Otherwise, the economic loss would be underestimated. The
authors should clarify this.

2. To cascade the impacts of working days lost into production supply chain, the per-
centage reductions in labor time loss were directly used as the percentage reductions
in industrial value added in the IO model. This should be based on the presumption
that labor inputs dominate the industrial values added, which might not be the case in
certain capital-intensive industry. The authors may need to clarify to what extent labor
inputs contribute to industrial values added in different sectors, and if not dominant,
how this assumption will bias their results.

Specific comments:

1. The Methods section should go before the Results section.

2. How was the direct economic loss calculated?

3. The Introduction only contains one citation. A plenty of sentences need references.
For example, "Serious air pollution in China has largely inspired epidemic studies that
examine ..."; "Existing epidemic studies simulate a exposure-response relationships
between Particulate Matter (PM) concentration levels and relative risks (RRs) for a
particular disease"; "while health costs assessments frequently stem from patients'

perspective at microeconomic level. . .”; “Inspired by our previous work. . .”, and etc.

4. Line 85, “Guangdong province. . ., where a substantial increase can be observed at 175 thousands compared with results in 2007”. Why did it increase? Additionally, "175 thousands" should be "175 thousand".

5. There're several sentences start with “I” such as sentences in Line 107, 121, 159, and etc, where “we” is expected.

6. Line 112, ” Primary industry includes agriculture and fishing suffered the economic loss at 19.12 billion Yuan”. The sentence needs to be rewritten.

7. Line 123, “The left-hand side shows the regional indirect economic loss while the right-hand side denotes the sources for these indirect economic loss. The proportion of regional indirect loss among regional total economic loss is displayed next to each region's name on the left-hand side”. Some of the information has been mentioned in the figure caption and thus don't need to be mentioned in the main text.

8. Line 355, “We referred to an integrated exposure-response (IER) model developed by Burnett et al (2014) . . .”. The reference was not found in the reference list. Also, the IER model in Burnett's study cannot be used to estimate hospital admissions and outpatient visits. need more accurate description here.

9. Line 370, “Then, the calculated RR was then converted into an attributable fraction (AF) in Eq.(2)”. Delete the second (or the first) “then”.

10. Line 385, “Î$\check{S}$: the parameter that describes the depth of the curve (Table SI-1 in Supplementary Information)”. There're two Table S1 in Supplementary Information.
* * *

---

## Referee Comment (RC2) · Anonymous Referee #2 · 16 Jul 2018

This study evaluates the economics of reduced labor time associated with pollution induced heath outcomes. The assessment framework is comprehensive and the manuscript is well organized. My major concern is the uncertainty of this assessment.

1. The PM2.5 concentration estimated by Geng et al. (2015) is based on satellite data. How are they compared with surface PM2.5 measurements?

2. The IER curve is applied in this study, but which version? IER parameters experienced significant changes over the past years since proposed, how that will change the results in this study?

3. The industrial labor time loss is the most uncertain part here. 250 working days lost are assumed without any references. These numbers can have large impacts on

the reported conclusions. It would be great if the authors could provide confidence intervals to address these uncertainty issues.

4. Method part should be moved before results.

---

## Referee Comment (RC3) · Anonymous Referee #1 · 28 Jul 2018

I recommend publication after some minor changes: 1. Reply to the authors' response to the specific comment NO.2: Thanks for explaining this. I believe other readers also want to have this information. Please provide it in the Supplement text. 2. Reply to the authors' response to the specific comment NO.6: "includes" should be "including".

---

## Author Comment (AC1) · 28 Jul 2018

Responses to Comments from R1

1. One methodological issue is the translation of mortality into working days lost. In section 4.4, this is described as "each death will result in a total 250 working days lost . . .". It confuses me why one death only results in one-year working days lost instead of years he supposes to work till retirement. We often consider the PM2.5-induced health impacts in a cumulative way, and that's what the disability-adjusted life year (DALY) stand for. DALYs represent the lost years of "healthy" life. DALYs for one premature death is determined as the standard life expectancy at age of death in years. If one death only accounted for one lost year of healthy life, then DALY would be significantly underestimated. It's rather the same to calculate lost years for working. One may argue that working days lost in years after 2012 don't account for the economic loss in 2012. Then mortality occurring before 2012 should be taken into account to comprehend the 2012 economic loss. Otherwise, the economic loss would be underestimated. The authors should clarify this.

Thank you for your constructive comments. We fully agree that air pollution-induced mortality and morbidity will result in long-lasting impacts that will definitely far exceed a single labour year loss or certain amount of working time loss. However, the focus of the paper and the proposed multi regional input-output (MRIO) model is the resulting health outcome and its impacts on labour time among all Chinese labourers during a single year of 2012. We acknowledge that compared with our proposed MRIO model, DALYs approach tends to provide more information on economic loss from the standpoint of each individual patient. However, we suggest that methods like DALYs and willingness-to-pay approach are likely to lose their sights on macroeconomic impacts which consider interdependencies among industries.

We fully agree that we need to better clarify these points and have done this by adding '*However, when perceiving unhealthy laborers as degradation in labor input, macroeconomic implications for production supply chains lack investigation. While traditional approaches for health costs estimates are able to provide more information on economic loss from a standpoint of individual patients, we suggest that they are likely to lose sights on the cascading effects due to labour time loss across interrelating industries.*' and '*In other words, it aims to investigate the overall economic loss resulting from health-induced labour time reduction among all Chinese labourers for a year of 2012.*' in the introduction section.

2. To cascade the impacts of working days lost into production supply chain, the percentage reductions in labor time loss were directly used as the percentage reductions in industrial value added in the IO model. This should be based on the presumption that labor inputs dominate the industrial values added, which might not be the case in certain capital-intensive industry. The authors may need to clarify to what extent labor inputs contribute to industrial values added in different sectors, and if not dominant, how this assumption will bias their results.

Thank you for your insightful comments. We fully agree that in reality, industries express different levels of dependencies on labour and capital. However, we used percentage reductions in labour time as a direct indicator for reductions in industrial value added because it is a fundamental assumption of production expansion path underlying input-output model. The model assumes that the same proportional increase in output can be only achieved by simultaneous increases in both capital and labour input. In other words, reduction in an input can directly constrain the growth in output.

We have clarified this assumption in Method section Industrial Labour Time Loss subsection as '*We need to clarify that the industries can express very different levels of dependencies on capital and labour in reality. However, percentage reductions in labour time were used as a direct indicator for percentage reduction in industrial value added due to the assumption of production expansion path underlying input-output model. An input-output model assumes that proportional increase in industrial output can be only achieved by simultaneous increases in both capital and labour, indicating that any reduction in an input can directly constrain the output growth in all industries.*'

Specific comments:
1. The Methods section should go before the Results section.

Thank you. We have made the change accordingly.

2. How was the direct economic loss calculated?

Thank you for the question.

For a demand-driven input-output model, the direct effects on the economy with respect to households are measured by the initial dollar value change of sector j's output that resulted from a change in dollar value of final demand. Thus, direct economic losses estimate the initial reductions in the dollar value of sector j's output caused by the decrease in its final demand. In addition, because:

**x= Lf and Δx= LΔf**

**L= (I-A)$^{-1}$= I+A+A$^2$+A$^3$+...**

the direct effects are associated with **I** and measured by the initial output change in sector j caused by the change in final demand of this sector.

At the same time, we refer to the accumulated effects caused by industrial inter-dependencies as indirect effects. They measure the reductions in dollar value among the outputs of other sectors caused by the reduced output of sector j, which is caused by a decrease in its final demand. The indirect effects are related to **A+A$^2$+A$^3$+…**

While a demand-driven input-output model suggests that production affects sectors that provide its primary inputs, our proposed supply-driven input-output model emphasises that production could also affect sectors that purchase its outputs as inputs in their production processes. It takes the form as:
**x' = v' (I-B)$^{-1}$**

when **G = (I-B)$^{-1}$**, equation becomes:

**x' = v' G**  (resulting gross output in a row view) or
**x = G' v**  (resulting gross output in a column view)

**G** and **G'** are the output/Ghosh inverse and the element g$_{ij}$ indicates the value of each unit of primary inputs in sector i that enters sector j. The supply-side approach assumes a fixed output distribution. Therefore, with fixed output coefficient b$_{ij}$, we can trace changes in sectoral gross outputs caused by changes in the amount of primary inputs using **Δx = Δv'G**. The supply-driven I-O model measures the direct economic losses as the initial dollar value decrease of sector j's outputs that resulted from a decrease in the dollar value of value added and are related to **I**, whereas the indirect economic losses refer to the accumulated reductions in dollar value among other sectors' outputs caused by the reduced outputs of sector j, which is caused by a decrease in its value added and are associated with **B+B$^2$+B$^3$+…**.

3. The Introduction only contains one citation. A plenty of sentences need references. For example, "Serious air pollution in China has largely inspired epidemic studies that examine ..."; "Existing epidemic studies simulate a exposure-response relationships between Particulate Matter (PM) concentration

levels and relative risks (RRs) for a particular disease"; "while health costs assessments frequently stem from patients' perspective at microeconomic level. . ."; "Inspired by our previous work. . .", and etc.

Thank you for your comments. We have added in references in both text and reference list to support our arguments. We have also arranged reference list in alphabetic order.

4. Line 85, "Guangdong province. . ., where a substantial increase can be observed at 175 thousands compared with results in 2007". Why did it increase? Additionally, "175 thousands" should be "175 thousand".

Thank you for your comments. We have changed 'thousands' into 'thousand'.

The reason behind the substantial rise in hospital admissions from its 2007 level is mainly because of a larger exposed population (from 94.49 million in 2007 to 105.94 million in 2012), despite sustained level in pollution concentrations.

5. There're several sentences start with "I" such as sentences in Line 107, 121, 159, and etc, where "we" is expected.

Thank you for your comments. We have changed them into 'We'.

6. Line 112, " Primary industry includes agriculture and fishing suffered the economic loss at 19.12 billion Yuan". The sentence needs to be rewritten.

Thank you for your comments. We have rewritten the sentence as '*Primary industry includes agriculture and fishing entail the economic loss at 19.12 billion Yuan*.'

7. Line 123, "The left-hand side shows the regional indirect economic loss while the right-hand side denotes the sources for these indirect economic loss. The proportion of regional indirect loss among regional total economic loss is displayed next to each region's name on the left-hand side". Some of the information has been mentioned in the figure caption and thus don't need to be mentioned in the main text.

Thank you for your comments. We have removed the repetitive sentences.

8. Line 355, "We referred to an integrated exposure-response (IER) model developed by Burnett et al (2014) . . .". The reference was not found in the reference list. Also, the IER model in Burnett's study cannot be used to estimate hospital admissions and outpatient visits. need more accurate description here.

Thank you for your comments. We have added Burnett et al (2014) into the reference list.

For air pollution-induced morbidity (hospital admissions and outpatient visits), we referenced the method used in Xia et al (2016) and Jiang et al (2015).

Specifically, we evaluated cardiovascular and respiratory hospital admissions and outpatient visits for all causes as the health endpoints of PM$_{2.5}$ air pollution. The log-linear response function was applied to estimate health outcomes, and the relative $RR$ for morbidity estimation was calculated by:

$RR = e^{\beta x}$

where $\beta$ is the parameter that describes the depth of the curve (SI-Table SI-1). They are the exposure-response coefficients which are used to quantify the relationship between different levels of PM$_{2.5}$ exposures and the resulting health effects. Then, the calculated RR were further converted into health outcome (hospital admission counts and outpatient visits) using the PAF formula.

9. Line 370, "Then, the calculated RR was then converted into an attributable fraction (AF) in Eq.(2)". Delete the second (or the first) "then".

Thank you for your comments. We have made the change accordingly.

10. Line 385, "ÎŠ: the parameter that describes the depth of the curve (Table SI-1 in Supplementary Information)". There're two Table S1 in Supplementary Information.

Thank you for your comments. We have corrected this in Supplementary Information.

---

## Author Comment (AC2) · 29 Jul 2018

Response to Comments from R2

This study evaluates the economics of reduced labor time associated with pollution induced heath outcomes. The assessment framework is comprehensive and the manuscript is well organized. My major concern is the uncertainty of this assessment.

1.  The PM2.5 concentration estimated by Geng et al. (2015) is based on satellite data. How are they compared with surface PM2.5 measurements?

Thank you for your question. We acknowledge that satellite AOD retrievals have regional biases compared with ground measurements and surface reflectance. To reduce such possible uncertainties, we employed the method from van Donkelaar et al (2010) to distinguish surface types using black-sky albedo and identify regional errors in AOD retrievals by extending biases calculated against ground measurements within a certain surface type. Specifically, we first identified four dominating surface types in China and used ground AOD measurements between 2006-12 to calculate monthly mean bias of satellite AOD and interpolated in each defined surface type. We excluded daily satellite AOD data with monthly bias larger than ±20%, which were further averaged to obtain the estimates of final long-term retrieval.

Van Donkelaar, Aaron, et al. "Global estimates of ambient fine particulate matter concentrations from satellite-based aerosol optical depth: development and application." *Environmental health perspectives* 118.6 (2010): 847.

2. The IER curve is applied in this study, but which version? IER parameters experienced significant changes over the past years since proposed, how that will change the results in this study?

Thank you for your comments. We referenced the IER functions developed by Burnett et al (2014). His proposed IER model incorporates data from cohort studies of ambient air pollution, and second-hand and active tobacco smoke to describe the concentration–response relationship throughout the full distribution of ambient PM2.5 concentrations, especially including the high levels in China. Therefore, we perceive this approach is suitable for estimating air pollution health impacts at high levels, such as the case in China, in the absence of epidemiological studies of the effects of long-term exposure to PM2.5. Given its applicability to a wide range of PM2.5 concentrations, the GBD also project employed these functions to estimate the global mortality due to ambient particulate and household air pollution in 2010 (Lim et al, 2012).

Burnett, Richard T., et al. "An integrated risk function for estimating the global burden of disease attributable to ambient fine particulate matter exposure." *Environmental health perspectives* 122.4 (2014): 397.

Lim, Stephen S., et al. "A comparative risk assessment of burden of disease and injury attributable to 67 risk factors and risk factor clusters in 21 regions, 1990–2010: a systematic analysis for the Global Burden of Disease Study 2010." *The lancet* 380.9859 (2012): 2224-2260.

3. The industrial labor time loss is the most uncertain part here. 250 working days lost are assumed without any references. These numbers can have large impacts on the reported conclusions. It would be great if the authors could provide confidence intervals to address these uncertainty issues.

Thank you for your comments. For the pollution-induced mortality, the 250 working day loss was inferred based on the business days calculator in China, which can be found on http://china.workingdays.org/EN

For pollution-induced morbidity, each cardiovascular admission will result in 11.9 working days lost while each respiratory admission causes 8.4 working days lost. We obtained the statistics from National Bureau of Statistics of China (2016). Meanwhile, we referenced Xia et al (2016) to provide a range for labor time loss estimation of outpatient visits due to data unavailability, which ranges from 2 to 4 hours per outpatient visit. We assumed each outpatient visits clinic once during the year.

We acknowledge the uncertainties involved in the evaluation. However, given the current data constraints, we feel that such assumption tends to provide a relatively conservative estimate regarding the disease induced labour time loss and the resulting economic impacts. We also provide sensitivity analysis for alternative hospital admission and outpatient time, indicating an upper and lower boundary for our estimates.

4. Method part should be moved before results.

Thank you for your comments. We have moved the Method section before Results section.

---

## Author Comment (AC3) · 30 Jul 2018

Response to Comments from R1 - 2 nd round
I recommend publication after some minor changes:

1. Reply to the authors' response to the specific comment NO.2: Thanks for explaining this. I believe other readers also want to have this information. Please provide it in the Supplement text.

Thank you for your comments. We have added these explanation to Method section as '*We need to clarify that the industries can express very different levels of dependencies on capital and labour in reality. However, percentage reductions in labour time were used as a direct indicator for percentage reduction in industrial value added due to the assumption of production expansion path underlying input-output model. An input-output model assumes that proportional increase in industrial output can be only achieved by simultaneous increases in both capital and labour, indicating that any reduction in an input can directly constrain the output growth in all industries.*' and to Supporting Information S1.3 A 'Circular Economy' in Input-Output Analysis as '*It worth noting that IO model possesses a fundamental assumption of production expansion path that assumes proportional increase in industrial output can be only achieved by simultaneous increases in both capital and labour, indicating that any reduction in an input can directly constrain the output growth in all industries.*'

2. 2. Reply to the authors' response to the specific comment NO.6: "includes" should be "including".

Thank you for your comments. We have corrected the word.

---

## Author Comment (AC4) · 30 Jul 2018

**Supplementary Information**

**S1. Selective Literature Review**

**S1.1 Exposure-Response Coefficients**

Particulate Matter (PM) affects human health most severely among all air pollutants and Fine Particle (PM$_{2.5}$) has substantially increased the numbers of mortality and morbidity (hospital admissions and outpatient visits) worldwide (Xu et al, 2000). The exposure-response coefficient is used to measure the quantitative relationship between PM exposure and its health outcomes. Xu et al (2000), Venners et al (2003) and Kan and Chen (2004) assessed the daily PM-induced mortality in Shenyang, Chongqing and Shanghai. Epidemic studies using exposure-response coefficients confirm increasing risks for mortality and morbidity of contaminant diseases at higher exposure (particulate concentration level).

**S1.2 Health Costs Assessment**

[revised manuscript text omitted]

---

## Author Comment (AC5) · 10 Aug 2018

[revised manuscript text omitted]

Miraglia, Simone Georges El Khouri, Paulo Hilário Nascimento Saldiva, and György Miklós Böhm. "An evaluation of air pollution health impacts and costs in São Paulo, Brazil." *Environmental management* 35.5, 667-676, (2005).

Mcghee, Sarah M., et al. "Cost of tobacco-related diseases, including passive smoking, in Hong Kong." *Tobacco control* 15.2, 125-130, (2006).

Miller, R. E. & P. D. Blair, 2009. Input-output analysis: foundations and extensions. Cambridge
University Press.

Meng J, Liu J, Xu Y, Tao S. Tracing Primary PM2.5 emissions via Chinese Supply Chains. *Environmental Research Letters* 10, 054005 (2015).

Meng J, Liu J, Xu Y, Guan D, Liu Z, Huang Y, Tao S. Globalization and pollution: tele-connecting local primary PM2.5 emissions to global consumption, *Pro. R. Soc. A. The Royal Society,* p 20160380 (2016).

Meng J, Liu J, Fan S, Kang C, Yi K, Cheng, Y, Shen, X, Tao S. Potential health benefits of controlling dust emissions in Beijing. *Environ. Pollut.* 213, 850-859 (2016).

National Bureau of Statistics of China (2013). National Statistical Yearbook 2013. [Online]. Available at http://www.stats.gov.cn/tjsj/ndsj/2013/indexch.htm Accessed 31/1/2017.

Santos JR, Haimes YY. 2004. Modeling the Demand Reduction Input-Output (I-O) Inoperability Due to Terrorism of Interconnected Infrastructures. *Risk Analysis* 24: 1437-51.

Steenge, A. E. & M. Bočkarjova, 2007. Thinking about imbalances in post-catastrophe economies: an input–output based proposition. Economic Systems Research 19(2): 205-223.

Sun J, Peng W. Domestic Interregional trade based on regional trade relations (基于区域贸易联系的国内区域贸易合作). *Social Science Research (社会科学研究)*. (2011). (in Chinese).

Venners, Scott A., et al. "Particulate matter, sulfur dioxide, and daily mortality in Chongqing, China."
*Environmental health perspectives* 111.4, 562 (2003).

Wong TW*, et al.* Air pollution and hospital admissions for respiratory and cardiovascular diseases in Hong Kong. *Occupational and environmental medicine* **56**, 679-683 (1999).

Wong C-M*, et al.* A tale of two cities: effects of air pollution on hospital admissions in Hong Kong and
London compared. *Environmental health perspectives* **110**, 67 (2002).

Wan Y, Yang H, Masui T. Air pollution-induced health impacts on the national economy of China: demonstration of a computable general equilibrium approach. *Reviews on environmental health* **20**, 119-140 (2004).

Wang H, Mullahy J. Willingness to pay for reducing fatal risk by improving air quality: a contingent valuation study in Chongqing, China. *Science of the Total Environment* **367**, 50-57 (2006).

Wang, X. J., et al. Air quality improvement estimation and assessment using contingent valuation
method, a case study in Beijing. *Environmental Monitoring and Assessment* 120.1-3: 153-168 (2006).

Wiedmann T, Minx J, Barrett J, Wackernagel M. Allocating ecological footprints to final consumption categories with input–output analysis. *Ecological economics* **56**, 28-48 (2006).

Xu Z, Yu D, Jing L, Xu X. Air pollution and daily mortality in Shenyang, China. *Archives of Environmental Health: An International Journal* **55**, 115-120 (2000).

Xia Y, Guan D, Jiang X, Peng L, Schroeder H, Zhang Q. Assessment of socioeconomic costs to China's air pollution. *Atmospheric Environment* 139 (2016): 147-56.

Xia, Yang, et al. "Assessment of the economic impacts of heat waves: A case study of Nanjing, China." *Journal of Cleaner Production* 171 (2018): 811-819.

Zeng X, Jiang Y. 2010. Evaluation of value of statistical life in health costs attributable to air pollution. *China Environmental Science* 30: 284-8.

.

---

## Author Response (AR1)

**Response to comments**

**Responses to RC1**

1. One methodological issue is the translation of mortality into working days lost. In section 4.4, this is described as "each death will result in a total 250 working days lost . . .". It confuses me why one death only results in one-year working days lost instead of years he supposes to work till retirement. We often consider the PM2.5-induced health impacts in a cumulative way, and that's what the disability-adjusted life year (DALY) stand for. DALYs represent the lost years of "healthy" life. DALYs for one premature death is determined as the standard life expectancy at age of death in years. If one death only accounted for one lost year of healthy life, then DALY would be significantly underestimated. It's rather the same to calculate lost years for working. One may argue that working days lost in years after 2012 don't account for the economic loss in 2012. Then mortality occurring before 2012 should be taken into account to comprehend the 2012 economic loss. Otherwise, the economic loss would be underestimated. The authors should clarify this.

Thank you for your constructive comments. We fully agree that air pollution-induced mortality and morbidity will result in long-lasting impacts that will definitely far exceed a single labour year loss or certain amount of working time loss. However, the focus of the paper and the proposed multi regional input-output (MRIO) model is the resulting health outcome and its impacts on labour time among all Chinese labourers during a single year of 2012. We acknowledge that compared with our proposed MRIO model, DALYs approach tends to provide more information on economic loss from the standpoint of each individual patient. However, we suggest that methods like DALYs and willingness-to-pay approach are likely to lose their sights on macroeconomic impacts which consider interdependencies among industries.

We fully agree that we need to better clarify these points and have done this by adding '*However, when perceiving unhealthy laborers as degradation in labor input, macroeconomic implications for production supply chains lack investigation. While traditional approaches for health costs estimates are able to provide more information on economic loss from a standpoint of individual patients, we suggest that they are likely to lose sights on the cascading effects due to labour time loss across interrelating industries.*' and '*In other words, it aims to investigate the overall economic loss resulting from health-induced labour time reduction among all Chinese labourers for a year of 2012.*' in the introduction section.

2. To cascade the impacts of working days lost into production supply chain, the percentage reductions in labor time loss were directly used as the percentage reductions in industrial value added in the IO model. This should be based on the presumption that labor inputs dominate the industrial values added, which might not be the case in certain capital-intensive industry. The authors may need to clarify to what extent labor inputs contribute to industrial values added in different sectors, and if not dominant, how this assumption will bias their results.

Thank you for your insightful comments. We fully agree that in reality, industries express different levels of dependencies on labour and capital. However, we used percentage reductions in labour time as a direct indicator for reductions in industrial value added because it is a fundamental assumption of production expansion path underlying input-output model. The model assumes that the same proportional increase in output can be only achieved by simultaneous increases in both capital and labour input. In other words, reduction in an input can directly constrain the growth in output.

We have clarified this assumption in Method section Industrial Labour Time Loss subsection as '*We need to clarify that the industries can express very different levels of dependencies on capital and labour in reality. However, percentage reductions in labour time were used as a direct indicator for percentage reduction in industrial value added due to the assumption of production expansion path underlying input-output model. An input-output model assumes that proportional increase in industrial output can be only achieved by simultaneous increases in both capital and labour, indicating that any reduction in an input can directly constrain the output growth in all industries.*'

Specific comments:
1. The Methods section should go before the Results

section. Thank you. We have made the change accordingly.

2. How was the direct economic loss calculated?

Thank you for the question.

For a demand-driven input-output model, the direct effects on the economy with respect to households are measured by the initial dollar value change of sector j's output that resulted from a change in dollar value of final demand. Thus, direct economic losses estimate the initial reductions in the dollar value

of sector j's output caused by the decrease in its final demand. In addition, because:

**x= Lf and Δx= LΔf**

**L= (I-A)$^{-1}$= I+A+A$^2$+A$^3$+...**

the direct effects are associated with **I** and measured by the initial output change in sector j caused by the change in final demand of this sector.

At the same time, we refer to the accumulated effects caused by industrial inter-dependencies as indirect effects. They measure the reductions in dollar value among the outputs of other sectors caused by the reduced output of sector j, which is caused by a decrease in its final demand. The indirect effects are related to **A+A$^2$+A$^3$+...**

While a demand-driven input-output model suggests that production affects sectors that provide its primary inputs, our proposed supply-driven input-output model emphasises that production could also affect sectors that purchase its outputs as inputs in their production processes. It takes the form as:
**x' = v' (I-B)$^{-1}$**

when **G = (I-B)$^{-1}$**, equation becomes:

**x' = v' G**  (resulting gross output in a row view) or
**x = G' v**  (resulting gross output in a column view)

**G** and **G'** are the output/Ghosh inverse and the element $g_{ij}$ indicates the value of each unit of primary inputs in sector i that enters sector j. The supply-side approach assumes a fixed output distribution. Therefore, with fixed output coefficient $b_{ij}$, we can trace changes in sectoral gross outputs caused by changes in the amount of primary inputs using **Δx = Δv'G**. The supply-driven I-O model measures the direct economic losses as the initial dollar value decrease of sector j's outputs that resulted from a decrease in the dollar value of value added and are related to **I**, whereas the indirect economic losses refer to the accumulated reductions in dollar value among other sectors' outputs caused by the reduced outputs of sector j, which is caused by a decrease in its value added and are associated with **B+B$^2$+B$^3$+...**.

3. The Introduction only contains one citation. A plenty of sentences need references. For example, "Serious air pollution in China has largely inspired epidemic studies that examine ..."; "Existing epidemic studies simulate a exposure-response relationships between Particulate Matter (PM) concentration

levels and relative risks (RRs) for a particular disease"; "while health costs assessments frequently stem from patients' perspective at microeconomic level. . ."; "Inspired by our previous work. . .", and etc.

Thank you for your comments. We have added in references in both text and reference list to support our arguments. We have also arranged reference list in alphabetic order.

4. Line 85, "Guangdong province. . ., where a substantial increase can be observed at 175 thousands compared with results in 2007". Why did it increase? Additionally, "175 thousands" should be "175 thousand".

Thank you for your comments. We have changed 'thousands' into 'thousand'.

The reason behind the substantial rise in hospital admissions from its 2007

level
is mainly because of a larger exposed population (from 94.49 million in 2007 to 105.94 million in 2012), despite sustained level in pollution concentrations.

5. There're several sentences start with "I" such as sentences in Line 107, 121, 159, and etc, where "we" is expected.

Thank you for your comments. We have changed them into 'We'.

6. Line 112, " Primary industry includes agriculture and fishing suffered the economic loss at 19.12 billion Yuan". The sentence needs to be rewritten.

Thank you for your comments. We have rewritten the sentence as '*Primary industry includes agriculture and fishing entail the economic loss at 19.12 billion Yuan.*'

7. Line 123, "The left-hand side shows the regional indirect economic loss while the right-hand side denotes the sources for these indirect economic loss. The proportion of regional indirect loss among regional total economic loss is displayed next to each region's name on the left-hand side". Some of the information has been mentioned in the figure caption and thus don't need to be mentioned in the main text.

Thank you for your comments. We have removed the repetitive sentences.

8. Line 355, "We referred to an integrated exposure-response (IER) model developed by Burnett et al (2014) . . .". The reference was not found in the reference list. Also, the IER model in Burnett's study cannot be used to estimate hospital admissions and outpatient visits. need more accurate description here.

Thank you for your comments. We have added Burnett et al (2014) into the reference list.

For air pollution-induced morbidity (hospital admissions and outpatient visits), we referenced the method used in Xia et al (2016) and Jiang et al (2015).

Specifically, we evaluated cardiovascular and respiratory hospital admissions and outpatient visits for all causes as the health endpoints of PM$_{2.5}$ air pollution. The log-linear response function was applied to estimate health outcomes, and the relative $RR$ for morbidity estimation was calculated by:

$RR = e^{\beta x}$

where $\beta$ is the parameter that describes the depth of the curve (SI-Table SI-1). They are the exposure-response coefficients which are used to quantify the relationship between different levels of PM$_{2.5}$ exposures and the resulting health effects. Then, the calculated RR were further converted into health outcome (hospital admission counts and outpatient visits) using the PAF formula.

9. Line 370, "Then, the calculated RR was then converted into an attributable fraction (AF) in Eq.(2)". Delete the second (or the first) "then".

Thank you for your comments. We have made the change accordingly.

10. Line 385, "ΊŠ: the parameter that describes the depth of the curve (Table SI-1 in Supplementary Information)". There're two Table S1 in Supplementary Information.

Thank you for your comments. We have corrected this in Supplementary
Information.

**Response to RC2**

This study evaluates the economics of reduced labor time associated with pollution induced heath outcomes. The assessment framework is comprehensive and the manuscript is well organized. My major concern is the uncertainty of this assessment.

1. The PM2.5 concentration estimated by Geng et al. (2015) is based on satellite data. How are they compared with surface PM2.5 measurements?

Thank you for your question. We acknowledge that satellite AOD retrievals have regional biases compared with ground measurements and surface reflectance. To reduce such possible uncertainties, we employed the method from van Donkelaar et al (2010) to distinguish surface types using black-sky albedo and identify regional errors in AOD retrievals by extending biases calculated against ground measurements within a certain surface type. Specifically, we first identified four dominating surface types in China and used ground AOD measurements between 2006-12 to calculate monthly mean bias of satellite AOD and interpolated in each defined surface type. We excluded daily satellite AOD data with monthly bias larger than ±20%, which were further averaged to obtain the estimates of final long-term retrieval.

Van Donkelaar, Aaron, et al. "Global estimates of ambient fine particulate matter concentrations from satellite-based aerosol optical depth: development and application." *Environmental health perspectives* 118.6 (2010): 847.

2. The IER curve is applied in this study, but which version? IER parameters experienced significant changes over the past years since proposed, how that will change the results in this study?

Thank you for your comments. We referenced the IER functions developed by Burnett et al (2014). His proposed IER model incorporates data from cohort studies of ambient air pollution, and second-hand and active tobacco smoke to describe the concentration−response relationship throughout the full distribution of ambient PM2.5 concentrations, especially including the high levels

in China. Therefore, we perceive this approach is suitable for estimating air pollution health impacts at high levels, such as the case in China, in the absence of epidemiological studies of the effects of long-term exposure to PM2.5. Given its applicability to a wide range of PM2.5 concentrations, the GBD also project employed these functions to estimate the global mortality due to ambient particulate and household air pollution in 2010 (Lim et al, 2012).

Burnett, Richard T., et al. "An integrated risk function for estimating the global burden of disease attributable to ambient fine particulate matter exposure." *Environmental health perspectives* 122.4 (2014): 397.

Lim, Stephen S., et al. "A comparative risk assessment of burden of disease and injury attributable to 67 risk factors and risk factor clusters in 21 regions, 1990–2010: a systematic analysis for the Global Burden of Disease Study 2010." *The lancet* 380.9859 (2012): 2224-2260.

3. The industrial labor time loss is the most uncertain part here. 250 working days lost are assumed without any references. These numbers can have large impacts on the reported conclusions. It would be great if the authors could provide confidence intervals to address these uncertainty issues.

Thank you for your comments. For the pollution-induced mortality, the 250 working day loss was inferred based on the business days calculator in China, which can be found on http://china.workingdays.org/EN

For pollution-induced morbidity, each cardiovascular admission will result in 11.9 working days lost while each respiratory admission causes 8.4 working days lost. We obtained the statistics from National Bureau of Statistics of China (2016). Meanwhile, we referenced Xia et al (2016) to provide a range for labor time loss estimation of outpatient visits due to data unavailability, which ranges from 2 to 4 hours per outpatient visit. We assumed each outpatient visits clinic once during the year.

We acknowledge the uncertainties involved in the evaluation. However, given the current data constraints, we feel that such assumption tends to provide a relatively conservative estimate regarding the disease induced labour time loss and the resulting economic impacts. We also provide sensitivity analysis for alternative hospital admission and outpatient time, indicating an upper and lower boundary for our estimates.

4. Method part should be moved before results.

Thank you for your comments. We have moved the Method section before Results section.

**Response to RC3**

I recommend publication after some minor changes:

1. Reply to the authors' response to the specific comment NO.2: Thanks for explaining this. I believe other readers also want to have this information. Please provide it in the Supplement text.

Thank you for your comments. We have added these explanation to Method section as '*We need to clarify that the industries can express very different levels of dependencies on capital and labour in reality. However, percentage reductions in labour time were used as a direct indicator for percentage reduction in industrial value added due to the assumption of production expansion path underlying input-output model. An input-output model assumes that proportional increase in industrial output can be only achieved by simultaneous increases in both capital and labour, indicating that any reduction in an input can directly constrain the output growth in all industries.*' and to Supporting Information S1.3
A 'Circular Economy' in Input-Output Analysis as '*It worth noting that IO model possesses a fundamental assumption of production expansion path that assumes proportional increase in industrial output can be only achieved by simultaneous increases in both capital and labour, indicating that any reduction in an input can directly constrain the output growth in all industries.*'

2. 2. Reply to the authors' response to the specific comment NO.6: "includes" should be "including".

Thank you for your comments. We have corrected the word.

**Assessment of Pollution-Health-Economics Nexus in China**

Yang Xia[1], Dabo Guan[1*], Jing Meng[2], Yuan Li[1], Yuli Shan[1]

[1] Water Security Research Centre, School of International Development, University of East Anglia, Norwich NR4 7TJ, UK
[2] Department of Politics and International Studies, University of Cambridge, Cambridge CB3 9DT, UK

* Correspondence email: Dabo.Guan@uea.ac.uk

**Abstract**

Serious haze can cause contaminant diseases that trigger productive labour time by raising mortality and morbidity rates in cardiovascular and respiratory diseases. Health studies rarely consider macroeconomic impacts of industrial interlinkages while disaster studies seldom involve air pollution and its health consequences. This study adopts a supply-driven input-output model to estimate the economic loss resulting from disease-induced working time reduction across 30 Chinese provinces in 2012 using the most updated Chinese Multiregional Input-Output Table. Results show total economic loss of 398.23 billion Yuan (~1% of China's GDP in 2012) with the majority comes from Eastern China and Mid-South. Total number of affected labourers amounts at 82.19 million. Cross-regional economic impact analysis indicates that Mid-South, North China and Eastern China entail the majority of regional indirect loss. Indeed, most indirect loss in North China, Northwest and Southwest can be attributed to Manufacturing and Energy in other regions while loss in Eastern China, Mid-South and Northeast largely originate from Coal and Mining in other regions. At the sub-industrial level, most inner-regional loss in North China and Northwest originate from Coal and Mining, in Eastern China and Southwest from Equipment and Energy, and in Mid-South from Metal and Non-metal. These findings highlight the potential role of geographical distance in regional interlinkages and regional heterogeneity in inner- and outer-regional loss due to distinctive regional economic structures and dependences between the North and South.

Keywords: China, 2012, PM$_{2.5}$ air pollution, Indirect economic loss, Health, Input-Output analysis

**1.Introduction**

Millions of people in China are currently breathing a toxic cocktail of chemicals, which has become one of the most serious topics in environmental issues in China by resulting in widespread environmental and health problems, including increasing risks for heart and respiratory diseases, stroke and lung cancer (LC) (Greenpeace, 2017). As air pollution has long-term health impacts that evolves gradually over time, understanding the health and socioeconomic impacts of China's air pollution requires continuous efforts.

Serious air pollution in China has largely inspired epidemic studies that examine specific health outcomes from air pollution, as well as health costs assessments that translate health outcomes into monetary loss (Xu et al, 2000; Venners et al, 2003; Kan and Chen, 2004). Existing epidemic studies simulate a exposure-response relationships between Particulate Matter (PM) concentration levels and relative risks (RRs) for a particular disease (see Xu et al, 2000; Venners et al, 2003) while health costs assessments frequently stem from patients' perspective at microeconomic level, by evaluating either their willingness-to-pay (WTP) for avoiding disease risk (see Wang and Mullahy, 2006; Wang et al, 2006) or the potentially productive years of life loss (PPYLL) (see Miraglia et al, 2005; Mcghee et al, 2006). However, when perceiving unhealthy laborers as degradation in labor input, macroeconomic implications for production supply chains lack investigation. While traditional approaches for health costs estimates are able to provide more information on economic loss from a standpoint of individual patients, we suggest that they are likely to lose sights on the cascading effects due to labour time loss across interrelating industries. Meanwhile, as the health effects of air pollution are built up slowly over time which implies the lasting nature of air pollution, it has been rarely studied in current disaster risk literature. Differing from rapid-onset disaster analysis (flood, hurricane, earthquake, etc) that normally reply on quantifying damages to physical capital, air pollution affect more human capital than physical capital and the resulting health impacts are relatively invisible and unmeasurable. As a result, linking PM concentrations with health endpoints and further with macroeconomic impacts require an interdisciplinary approach that integrates all the three elements into one. Inspired by our previous work on the socioeconomic impacts of China's air pollution in 2007 (Xia et al, 2016), this paper applies the similar approach to China's air pollution in 2012 and also examines the cross-regional economic impacts in order to underline the important role of indirect economic loss for the year 2012. In other words, it aims to investigate the overall economic loss resulting from health-induced labour time reduction among all Chinese labourers for a year of 2012. Given that the majority of economic loss originate from secondary industry, this paper also specifically analyze the key sectors in secondary industry that account for the greatest proportions in both direct and indirect economic loss in each great region in China. By doing so, future policymakers and researchers could obtain an alternative macroeconomic tool to better conduct cost-benefit analysis in any environmental or climate change related policy design, and to comprehend health costs studies in its macroeconomic side.

**2.Methods**

**2.1.  Methodological Framework**

[Figure]

Emission Inventory

*Air quality simulation models*

PM$_{2.5}$ Concentration Levels

**Environmental Studies**

*Exposure-response relationship*    *Exposure-response relationship*

PM$_{2.5}$-induced Mortality Risks    PM$_{2.5}$-induced Morbidity Risks

*Population Attributable Fraction (PAF)*    *Population Attributable Fraction (PAF)*

Excess Numbers of PM$_{2.5}$-induced Deaths, Admissions and Outpatient Visits

**Health Studies**

*Labour-population ratio*

Counts of PM$_{2.5}$-induced Deaths, Admissions and Outpatient Visits among Labour

*Average productive time loss for each health endpoint*

Industrial Productive Labour Time Loss

*Compare with original industrial productive time*

Percentage Reduced in Industrial Productive Labour Time

**Industrial Impact Analysis**

*Productive time as key to industrial value-added*

Percentage Reduced in Industrial Value-added

*A supply-driven input-output model*

Cascading Indirect Loss Along Production Supply Chains

*Multi-regional economic analysis*    *Multi-regional economic analysis*

Macroeconomic Implications (Direct + Indirect Economic Loss)    Cross-regional Economic Impact Analysis

**Macroeconomic Analysis**

*Figure                                                                                          1*

[revised manuscript text omitted]

$$x' = v' \,(I-B)^{-1} \qquad\qquad \text{(Eq.8a)}$$

$$x' = v' \, G, \; G = (I-B)^{-1} \qquad\qquad \text{(Eq.8b)}$$

**B**: the allocation coefficient (direct-output coefficient), where $b_{ij} = z_{ij} / x_i$. It refers to the distribution of sector i's outputs in sector j
**v**: matrix of industrial value added, including capital and labour input
**G**: the Ghosh inverse matrix, which measures the economic impacts of changes in a sector's value added on other sectors' output level

**3.Results**

**3.1 Total Number of Affected Labor and Total Economic Loss**

Firstly, regarding the total number of affected labour and total economic loss, the total economic loss resulting from $PM_{2.5}$-induced health outcomes in China 2012 is 398.23 billion Yuan, which corresponds to almost 1% of national GDP in 2012. The total number of affected labour in China is

0.80 million for PM$_{2.5}$[1]-induced mortality, 2.22 million for PM$_{2.5}$-induced hospital admissions and 79.17 million for PM$_{2.5}$-induced outpatient visits (*Fig.2*). *Figure 1*). *Figure 1*2 presents the provincial counts of PM$_{2.5}$-induced mortality, hospital admissions, outpatient visits and economic loss with least severe and most severe situation shown from green to red. For total populations of PM$_{2.5}$-induced mortality and morbidity, among 30 provinces, Henan and Shangdong province have the largest total counts of PM$_{2.5}$-induced mortality and morbidity, which is consistent with the findings in 2007 study (Xia et al, 2016). Guangdong province has the greatest counts of PM$_{2.5}$-induced hospital admissions at 291 thousands, where a substantial increase can be observed at 175 thousand compared with results in 2007. It almost doubles its provincial count of outpatient visits and triples its mortality counts. Meanwhile, increase can be observed in both counts for Northwest region, including Shanxi, Gansu, Qinghai, Ningxia and Xinjiang provinces. Specifically, the count of hospital admissions in Shanxi province in 2012, 100 thousands, also doubles that of 2007, which was at 50 thousands. Even sharper increase of admission counts can be seen in Xinjiang province, where the number is almost 7 times of that from 2007.

[Figure]

*Figure 1*2. *Provincial Countscounts of PM$_{2.5}$-induced Mortality, Hospitalmortality, hospital Admissions, Outpatient Visitsoutpatient visits and Economic Losseconomic loss in China, 2012*
* * *
[1] PM$_{2.5}$ is also known as fine particulate matters that have a diameter of less than 2.5 micrometers.

[revised manuscript text omitted]

Miraglia, Simone Georges El Khouri, Paulo Hilário Nascimento Saldiva, and György Miklós Böhm. "An evaluation of air pollution health impacts and costs in São Paulo, Brazil." *Environmental management* 35.5, 667-676, (2005).

Mcghee, Sarah M., et al. "Cost of tobacco-related diseases, including passive smoking, in Hong Kong." *Tobacco control*15.2, 125-130, (2006).

Miller, R. E. & P. D. Blair, 2009. Input-output analysis: foundations and extensions. Cambridge University Press.

Meng J, Liu J, Xu Y, Tao S. Tracing Primary PM2.5 emissions via Chinese Supply Chains. *Environmental Research Letters* 10, 054005 (2015).

Meng J, Liu J, Xu Y, Guan D, Liu Z, Huang Y, Tao S. Globalization and pollution: tele-connecting local primary PM2.5 emissions to global consumption, *Pro. R. Soc. A. The Royal Society,* p 20160380 (2016).

Meng J, Liu J, Fan S, Kang C, Yi K, Cheng, Y, Shen, X, Tao S. Potential health benefits of controlling dust emissions in Beijing. *Environ. Pollut.* 213, 850-859 (2016).

National Bureau of Statistics of China (2013). National Statistical Yearbook 2013. [Online]. Available at http://www.stats.gov.cn/tjsj/ndsj/2013/indexch.htm Accessed 31/1/2017.

Santos JR, Haimes YY. 2004. Modeling the Demand Reduction Input-Output (I-O) Inoperability Due to Terrorism of Interconnected Infrastructures. *Risk Analysis* 24: 1437-51.

Steenge, A. E. & M. Bočkarjova, 2007. Thinking about imbalances in post-catastrophe economies: an input–output based proposition. Economic Systems Research 19(2): 205-223.

Sun J, Peng W. Domestic Interregional trade based on regional trade relations (基于区域贸易联系的国内区域贸易合作). *Social Science Research (社会科学研究).* (2011). (in Chinese).

Venners, Scott A., et al. "Particulate matter, sulfur dioxide, and daily mortality in Chongqing, China." *Environmental health perspectives* 111.4, 562 (2003).

Wong TW*, et al.* Air pollution and hospital admissions for respiratory and cardiovascular diseases in Hong Kong. *Occupational and environmental medicine* **56**, 679-683 (1999).

Wong C-M*, et al.* A tale of two cities: effects of air pollution on hospital admissions in Hong Kong and London compared. *Environmental health perspectives* **110**, 67 (2002).

Wan Y, Yang H, Masui T. Air pollution-induced health impacts on the national economy of China: demonstration of a computable general equilibrium approach. *Reviews on environmental health* **20**, 119-140 (2004).

Wang H, Mullahy J. Willingness to pay for reducing fatal risk by improving air quality: a contingent valuation study in Chongqing, China. *Science of the Total Environment* **367**, 50-57 (2006).

Wang, X. J., et al. Air quality improvement estimation and assessment using contingent valuation method, a case study in Beijing. *Environmental Monitoring and Assessment* 120.1-3: 153-168 (2006).

Wiedmann T, Minx J, Barrett J, Wackernagel M. Allocating ecological footprints to final consumption categories with input–output analysis. *Ecological economics* **56**, 28-48 (2006).

Xu Z, Yu D, Jing L, Xu X. Air pollution and daily mortality in Shenyang, China. *Archives of Environmental Health: An International Journal* **55**, 115-120 (2000).

Xia Y, Guan D, Jiang X, Peng L, Schroeder H, Zhang Q. Assessment of socioeconomic costs to China's air pollution. *Atmospheric Environment* 139 (2016): 147-56.

Xia, Yang, et al. "Assessment of the economic impacts of heat waves: A case study of Nanjing, China." *Journal of Cleaner Production* 171 (2018): 811-819.

Zeng X, Jiang Y. 2010. Evaluation of value of statistical life in health costs attributable to air pollution. *China Environmental Science* 30: 284-8.

.